# Can assimilation of satellite observations improve subsurface biological properties in a numerical model? A case study for the Gulf of Mexico

**Bin Wang**[1]**, Katja Fennel**[1]**, and Liuqian Yu**[1,2]

[1]Department of Oceanography, Dalhousie University, Halifax, Nova Scotia, Canada
[2]Department of Ocean Science, The Hong Kong University of Science and Technology, Kowloon, Hong Kong

**Correspondence:** Bin Wang (Bin.Wang@dal.ca)

**Abstract.** Given current threats to ocean ecosystem health, there is a growing demand for accurate biogeochemical hindcasts, nowcasts, and predictions. Provision of such products requires data assimilation, i.e., a comprehensive strategy for incorporating observations into biogeochemical models, but current data streams of biogeochemical observations are generally considered insufficient for the operational provision of such products. This study investigates to what degree the assimilation of satellite observations in combination with *a priori* model calibration by sparse BGC-Argo profiles can improve subsurface biogeochemical properties. The multivariate Deterministic Ensemble Kalman Filter (DEnKF) has been implemented to assimilate physical and biological observations into a three-dimensional coupled physical-biogeochemical model, of which the biogeochemical component has been calibrated by BGC-Argo floats data for the Gulf of Mexico. Specifically, observations of sea surface height, sea surface temperature, and surface chlorophyll were assimilated, and profiles of both physical and biological variables were updated based on the surface information. We assessed whether this leads to improved subsurface distributions, especially of biological properties, using observations from five BGC-Argo floats that were not assimilated. An alternative light parameterization that was tuned *a priori* using BGC-Argo observations was also applied to test the sensitivity of data assimilation impact on subsurface biological properties. Results show that assimilation of the satellite data improves model representation of major circulation features, which translate into improved three-dimensional distributions of temperature and salinity. The multivariate assimilation also improves the agreement of subsurface nitrate through its tight correlation with temperature, but the improvements in subsurface chlorophyll were modest initially

due to suboptimal choices of the model's optical module. Repeating the assimilation run by using the alternative light parameterization greatly improved the subsurface distribution of chlorophyll. Therefore, even sparse BGC-Argo observations can provide substantial benefits to biogeochemical prediction by enabling *a priori* model tuning. Given that, so far, the abundance of BGC-Argo profiles in the Gulf of Mexico and elsewhere is insufficient for sequential assimilation, updating 3D biological properties in a model that has been well calibrated is an intermediate step toward full assimilation of the new data types.

## 1 Introduction

Given the multiple and increasing pressures of ocean warming, acidification, deoxygenation, and changes in primary productivity on ocean ecosystem health, accurate model simulations are urgently needed to assess past and current states of marine ecosystems, forecast future trends, and predict the ocean's response to different scenarios of climate change and management policies. In practice, numerical models are imperfect representations of the natural system and their accuracy is limited by many factors including insufficient model resolution, inaccuracies in discretization schemes and model formulations, parameterization of unresolved processes, and uncertainties in model inputs. Data assimilation is a practical approach used to compensate for these model deficiencies. It is a statistical method to interpolate and extrapolate the sparse observations into the regular model space in a dynamically consistent way. Its success critically depends on well-resolved observations. While any practice to constrain a

model by observations can be referred to as data assimilation, in this manuscript we specifically refer to state estimation, i.e., sequential updates of the model state.

Data assimilation is well developed in physical oceanography (Edwards et al., 2015) but less mature in biogeochemical ocean modelling, largely due to insufficient observations (Fennel et al., 2019). Thus far, satellite data of ocean color (e.g. chlorophyll) have been the major source of observations to be assimilated (e.g. Ford and Barciela, 2017; Gregg, 2008; Hu et al., 2012; Mattern et al., 2013; Pradhan et al., 2019; Teruzzi et al., 2018) because of their relatively high resolution and routine availability. More recent advances have focused on the incorporation of other satellite-derived products including size-fractionated chlorophyll (e.g. Ciavatta et al., 2018, 2019; Pradhan et al., 2020; Skákala et al., 2018) and optical properties (e.g. Ciavatta et al., 2014; Gregg and Rousseaux, 2017; Jones et al., 2016; Shulman et al., 2013; Skákala et al., 2020). However, these measurements are limited to the surface ocean and provide little information about the subsurface and ocean interior. In addition, it has been acknowledged that assimilating satellite data of ocean color often fails to improve and even degrades simulation of unobserved biological variables (Ciavatta et al., 2018; Fontana et al., 2013; Ford and Barciela, 2017; Skákala et al., 2018; Teruzzi et al., 2018). Problems also remain in accounting for the co-dependencies or covariances between biological variables. For instance, Fontana et al. (2013) found subsurface nitrate was barely impacted by assimilating the satellite surface chlorophyll because of its weak correlations with surface chlorophyll. Although BGC-Argo floats may ultimately provide us with abundant subsurface measurements of multiple key biogeochemical properties (Biogeochemical-Argo Planning Group, 2016; Chai et al., 2020; Roemmich et al., 2019), the profiling observations will likely remain insufficient for three-dimensional data assimilation for a number of years, making satellite data the main observation streams for sequential data assimilation in biogeochemical models (Ford, 2021).

The insufficient availability of subsurface and interior ocean biogeochemical observations is not only reflected in the immaturity of biogeochemical data assimilation but also its skill assessment. When compared with the surface, the subsurface has received less attention in skill assessments of biogeochemical data assimilation systems. Although there have been studies that compared vertical structures with in-situ observations and/or climatological datasets (e.g. Fontana et al., 2013; Ford and Barciela, 2017; Mattern et al., 2017; Ourmières et al., 2009; Teruzzi et al., 2014), these validations were often limited to low spatio-temporal resolution. The recent growth of autonomous observation systems, esp. BGC-Argo floats and gliders, make it possible to evaluate biogeochemical data assimilation systems below the surface in high resolution (e.g. Cossarini et al., 2019; Salon et al., 2019; Skákala et al., 2021; Verdy and Mazloff, 2017).

Finally, since physical processes affect biological properties through advection and diffusion of biological tracers as well as some temperature-dependent biological activities (e.g. phytoplankton growth), deficiencies in biological models can arise from imperfect simulation of the physics (Doney, 1999; Doney et al., 2004; Oschlies and Garçon, 1999). Although there have been studies demonstrating a positive effect of physical data assimilation on biological properties (Fiechter et al., 2011; Ourmières et al., 2009), often this approach degrades biological distributions because of elevated vertical velocities and violation of consistency between physical and biological properties (Anderson et al., 2000; Raghukumar et al., 2015; Yu et al., 2018). To address these issues, joint assimilation of physical and biological observations (Song et al., 2016a, b) or multivariate updates based on the cross-covariances between physical and biological properties (Goodliff et al., 2019; Yu et al., 2018) have been suggested.

In this study, a multivariate physical-biological data assimilation scheme is applied to a coupled physical-biological model in the Gulf of Mexico. The rationale for choosing the Gulf of Mexico is that the dominant circulation, including the Loop Current and its associated mesoscale eddies, is stochastic and can influence the subsurface biological distributions, e.g. deep chlorophyll maximum (Fommervault et al., 2017). In addition, we test how data assimilation impacts depend on model calibration when using two alternative light parameterizations. By comparing forecast results from the assimilative model with independent observations from five BGC-Argo floats which are not assimilated but used in *a priori* tuning of the biogeochemical model, we rigorously evaluate whether the main biological observation stream (satellite estimates of surface chlorophyll) in combination with physical observations (satellite estimates of sea surface height and sea surface temperature) can inform the 3D ocean distributions in high spatial and temporal resolution.

## 2 Tools and methods

### 2.1 Coupled physical and biological model

The coupled physical and biological model used in this study is based on the Regional Ocean Modeling System (ROMS, Haidvogel et al., 2008) and configured in the Gulf of Mexico (red rectangle in Fig. 1 shows the model domain) with a horizontal resolution of ∼5 km and 36 vertical sigma levels (Wang et al., 2020; Yu et al., 2019). The model used a Multidimensional Positive Definitive Advection Transport Algorithm (MPDATA Smolarkiewicz and Margolin, 1998) to solve the horizontal and vertical advection of tracers, a Smagorinsky-type formula (Smagorinsky, 1963) to parameterize horizontal viscosity and diffusivity, and the Mellor-Yamada 2.5-level closure scheme (Mellor and Yamada, 1982) to calculate the vertical turbulent mix-

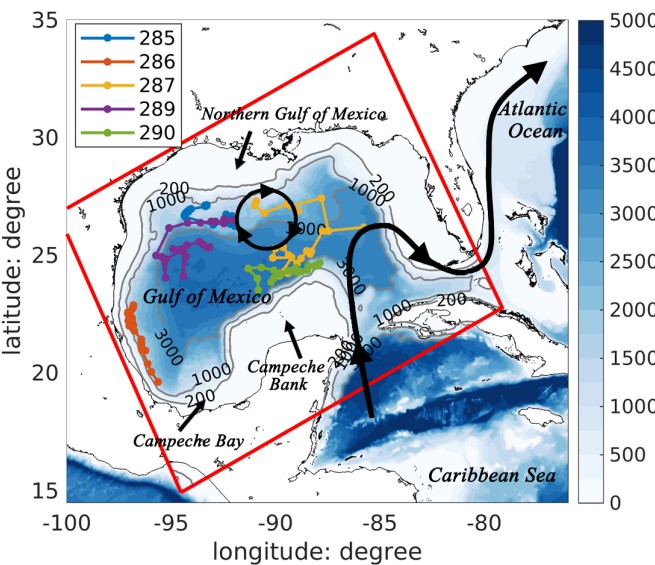

**Figure 1.** Bathymetric map of the Gulf of Mexico with a schematic pattern of Loop Current (black curve with arrows) and Loop Current Eddies (black circle with arrows). Trajectories of five BGC-Argo floats (colored lines) in 2015 were also shown in the figure. The model domain is represented by the red rectangle.

ing. Atmospheric forcing is provided by the European Centre for Medium-Range Weather Forecast ERA-Interim product (ECMWF reanalysis, Dee et al., 2011) with a horizontal resolution of 1/8° (approximately 12 km×14 km) to calculate the surface wind stress as well as the net heat fluxes and freshwater fluxes.

The biological model uses a nitrogen-based model (Fennel et al., 2006) to simulate transportation and transformation of seven pelagic variables, i.e. nitrate (NO3), ammonium (NH4), chlorophyll (Chl), phytoplankton (Phy), zooplankton (Zoo), small detritus (SDet), and large detritus (LDet). As a separate state variable, chlorophyll accounts for photoacclimation based on Geider et al. (1997). In our coupled model, the biological tracers are advected and diffused as part of the 3D circulation but provide no feedback to the physical model. Biological parameters are from the parameter optimization study of Wang et al. (2020) except that the half-saturation constant of nitrate was subjectively re-tuned based on the BGC-Argo floats data from 0.5 mmol N m$^{-3}$ to about 1.4 mmol N m$^{-3}$ because the previous model underestimated the nitrate in the euphotic zone.

The coupled model receives freshwater and nutrients inputs from the Mississippi-Atchafalaya river systems which are specified by the daily measurements from the US Geological Survey river gauges and those from other major rivers which utilize the climatological estimates (Xue et al., 2013). To ensure a dynamically consistent biological field, a one-year spin-up is performed in 2014 where the physical model is initialized from the output of the 1/12° data-assimilative

global HYCOM/NCODA (Chassignet et al., 2005) and the biological model starts from a regressed 3D field of nitrate based on its climatological relationship with temperature (see Fig. S1). A semi-prognostic method is used during the spin-up period to reduce model drift by replacing model density with a linear combination of model and climatological density fields when calculating the horizontal pressure gradient (Greatbatch et al., 2004; Sheng et al., 2001). After the spin-up, experiments are performed for a year from January 2015 to December 2015.

## 2.2 Data assimilation technique

In this study, the data assimilation scheme uses the deterministic formulation of the Ensemble Kalman Filter (DEnKF) which was first introduced by Sakov and Oke (2008). The approach consists of two steps: 1) the forecast step in which an ensemble of state variables is integrated forward in time by the model, and 2) the analysis step in which observations are assimilated to update the forecasted ensemble following the Kalman Filter equations

$$x^a = x^f + K(d - Hx^f), \tag{1}$$
$$K = P^f H^T (HP^f H^T + R)^{-1} \tag{2}$$

where $x$ represents the model state estimate, $d$ represents the available observations, $H$ represents the observation operator mapping the model state onto observations, and $K$ represents the Kalman gain matrix which is determined by the model error matrix $P$ and observation error matrix $R$ (Eq. 2). Superscripts $a$ and $f$ represent analysis (i.e. updated) and forecast (i.e. prior to the update) estimates, and $T$ represents the matrix transpose. Unlike the original stochastic EnKF which updates each ensemble member with perturbed observations, the DEnKF updates ensemble mean ($\overline{x}$) and anomalies ($A = x - \overline{x}$) separately without perturbing observations, i.e. the former is updated as in Eq. 1 while the latter is updated by

$$A^a = A^f - \frac{1}{2} KHA^f \tag{3}$$

More details of the DEnKF can be referred to Sakov and Oke (2008) and Yu et al. (2018).

The data assimilation framework and configurations are the same as in Yu et al. (2019) where twin experiments were performed in the same model domain. In this study, we extend the work to jointly assimilate the physical and biological observations into a coupled model. For the sake of keeping our data assimilation experiments computationally affordable, we chose an ensemble size of 20 which has been successfully used in previous studies including an idealized channel (Yu et al., 2018), the Middle Atlantic Bight (Hu et al., 2012; Mattern et al., 2013), and the Gulf of Mexico (Yu et al., 2019). Spurious correlations, which can arise with relatively small ensembles, are avoided here by applying a distance-based localization with a radius of 50 km (Evensen,

2003). Vertical localization is not applied. Ensemble anomalies are inflated by 1.05 in each update step to account for unrepresented sources of model uncertainty (Anderson and Anderson, 1999). Values of the localization radius and inflation factor were determined in Yu et al. (2019).

In order to account for uncertainties in the model's initial, boundary and atmospheric forcing conditions and biological parameters, the ensemble is initialized from 20 different daily outputs, centered on the initial date of 1 January 2015, from a previous deterministic model simulation (as described above in Section 2.1) and is forced by open boundary conditions which are lagged by up to $\pm10$ days for the different ensemble members. Furthermore, each ensemble member is forced by a perturbed version of the wind forcing. Specifically, the wind forcing from the deterministic run is decomposed into empirical orthogonal functions (EOFs) and then the first 4 EOFs are perturbed by multiplying random numbers with zero mean and variance of 1 as in Li et al. (2016) and Thacker et al. (2012). In addition, four sensitive biological parameters, namely the mortality rate of phytoplankton, the maximum ratio of chlorophyll to carbon, the grazing rate of zooplankton, and the growth rate of phytoplankton at 0 °C, were identified in sensitivity experiments. Specifically, a 1D version of model, described in Wang et al. (2020) was run multiple times while incrementally perturbing one parameter at a time by factors ranging from 0.25 to 1.75 with an increment of 0.25. The four sensitive parameters were selected based on the normalized absolute differences between the perturbed and unperturbed run. In the data assimilation experiments, these parameters are subject to a Gaussian perturbation with a relative variance of 75% but they are not updated. The parameters are resampled from their distributions before each forecast step to prevent some extreme parameter values being used throughout the whole data assimilation experiment.

## 2.3 Observations

In this study, physical and biological observations are jointly assimilated to constrain the coupled model. The observations assimilated include sea surface height (SSH), sea surface temperature (SST), Argo T-S profiles, and satellite estimates of surface chlorophyll.

The SSH observations for assimilation are obtained by adding the $1/4°$ mapped Sea level anomaly (SLA) from Archiving Validation and Interpretation of Satellite Oceanographic Data (AVISO) to a mean dynamic topography (MDT) from Rio et al. (2013), and are adjusted by removing the spatially averaged mismatches between assimilated and forecasted SSH to account for differences in reference time between the SLA data (1993-2012) and our coupled model (2015) (Haines et al., 2011; Song et al., 2016b; Xu et al., 2012). This is equivalent to assimilating the SSH gradient into the model, as it is the only dynamically meaningful quantity for driving the geostrophic component of ocean currents and adjusting subsurface thermohaline structures. The SST observations are Advanced Very High Resolution Radiometer (AVHRR, Martin et al., 2012) product with a horizontal resolution of $0.01°$. Observation errors are specified as 0.02 m for SSH and $0.3°C$ for SST (Song et al., 2016b; Yu et al., 2018, 2019).

The surface chlorophyll is provided by the Ocean-Colour Climate Change Initiative project (OC-CCI, Sathyendranath et al., 2018) at a daily frequency with a spatial resolution of $1/24°$. However, for the daily chlorophyll field, a large portion of data can be missing due to cloud cover and inter-orbit gaps. In 2015 for the Gulf of Mexico, the spatial coverage of surface chlorophyll varies from 0 to 63% with a mean coverage of $9.5\pm9.0\%$. Hence, to increase the availability of observations, an asynchronous data assimilation method (Sakov et al., 2010) is applied so that not only the daily records of surface chlorophyll at the date of update but also the daily records within the preceding 7 days are assimilated. Errors associated with the surface chlorophyll are set to be 35% of the measured concentrations, which has been commonly used in previous applications (e.g. Fontana et al., 2013; Ford, 2021; Ford and Barciela, 2017; Hu et al., 2012; Mattern et al., 2017; Santana-Falcón et al., 2020; Song et al., 2016b; Yu et al., 2018). In this study, the update is performed on actual chlorophyll concentrations because our prior tests showed that it outperforms assimilating log-chlorophyll in the open Gulf (with depth >1000 m). There are previous examples where the actual chlorophyll values have been assimilated successfully (e.g. Hu et al., 2012; Yu et al., 2018) although we note that assimilating the actual chlorophyll values is theoretically suboptimal because of their non-Gaussian distribution.

Profiling observations are from the International Argo project (hereafter referred to as Argo floats) and five BGC-Argo floats which were funded by the Bureau of Ocean Energy Management (hereafter referred to as BOEM floats). In 2015, the Argo floats provided nearly 800 T-S profiles extending from the surface to 2,000 m depth in the Gulf of Mexico. These are treated either as independent observations for model skill assessment or, in the DAargo experiment (see Section 2.4), assimilated with uncertainties being $0.3°C$ for temperature and 0.01 for salinity. The BOEM floats collected more than 500 profiles of temperature, salinity, chlorophyll, and backscatter at a bi-weekly frequency from 2011 to 2015, 114 profiles of which were collected in 2015 (see Fig. 1 for their locations) and are used as independent observations. Backscatter is converted into phytoplankton and particulate organic carbon (POC) concentrations following Wang et al. (2020). In the absence of direct measurements for nitrate, we estimate it along the BOEM float trajectories based on their climatological relationship with temperature (Fig. S1).

## 2.4 Simulation strategy

We performed five 1-year simulations in 2015. The first one is a deterministic model simulation without data assimilation (henceforth referred to as **Free** simulation). The second one is an ensemble run assimilating satellite data (SSH, SST, and satellite surface chlorophyll) only (henceforth **DAsat**), and the third one is an ensemble run assimilating Argo T-S profiles in addition to satellite data (henceforth **DAargo**). The light attenuation module ($Att = 0.04 + 0.025 \times chl$) used in these three simulations are from literatures (e.g. Fennel et al., 2006, 2011) , by which the light attenuation coefficient, $Att$, is strongly determined by water depth and not very sensitive to chlorophyll concentrations. The Free run and DAsat run are repeated by using an alternative light parametrization (henceforth referred to as **Free-alt** and **DAsat-alt** simulations, respectively) to evaluate its effect on the data-assimilation impact on subsurface biological properties. This alternative light parameterization ($Att = 0.027 + 0.075 \times chl^{1.2}$) is subjectively tuned based on the BGC-Argo observations and emphasizes the self-shading effect of chlorophyll on light attenuation.

A two-step update is used on a weekly data assimilation cycle in the assimilative experiments, where the physical observations are first assimilated to update both physical and biological state variables through the multivariate covariance, and chlorophyll observations are assimilated next to update only biological state variables. Although the DEnKF can update all state variables based on their cross-covariance, we limit updates to two physical variables (temperature and salinity) and four biological variables (nitrate, chlorophyll, phytoplankton, and zooplankton) which are key to the coupled physical-biogeochemical system. As the circulation features in the open Gulf (the Loop Current and its associated mesoscale eddies) are primarily in geostrophic balance, an update of temperature and salinity can improve three-dimensional circulation features in large scales effectively, as shown in the twin experiments in Yu et al. (2019). All these state variables are updated throughout the whole water column while other variables are adjusted by internal model dynamics.

To evaluate the prediction skill, we calculate the root-mean-square-errors (RMSE), the bias, and the correlation coefficient (Corr) of the model forecast ($M$) with respect to assimilated and independent observations ($O$):

$$RMSE = \sqrt{\frac{1}{N}\sum(M-O)^2}, \tag{4}$$

$$bias = \frac{1}{N}\sum(M-O) \tag{5}$$

where $N$ represents the number of model-data pairs available. To account for the overestimation of nitrate in warm waters which typically occurs in the euphotic zone (Fig. S1), an unbiased root-mean-square-error (unbiased RMSE) is used to quantify the model-data misfit of nitrate.

$$unbiased\ RMSE = \sqrt{\frac{1}{N}\sum(M-O-bias)^2} \tag{6}$$

## 3 Results

### 3.1 Assimilation impacts on physical properties

As the biological model provides no feedback to the physical model, the alternative light parameterization does not affect physical properties. The physical results from Free-alt and DAsat-alt runs are thus not displayed in this section.

The dominant circulation features in the Gulf of Mexico, the Loop Current and Loop Current Eddies, are assessed by comparing their fronts, defined here as the 10-cm SSH contour, from satellite data, the free run, and two data-assimilative runs (i.e. the DAsat and DAargo runs). In the first two months, all model estimates of the Loop Current are different from satellite observations due to the influence of initial conditions (Fig. 2). After March, the SSH field shows a similar northward and westward extension of the Loop Current intrusion between two assimilative runs and satellite observations, but large deviations from observations remain in the Free run. In addition, all estimates except for the Free run well reproduce the satellite-observed timing of eddy shedding as well as the size, shape, and position of Loop Current eddies.

For a more quantitative assessment, the daily output of SSH and SST fields from the three runs are compared with the satellite estimates. The spatial distribution of RMSE from the Free run and the RMSE changes in two data-assimilative runs are shown in Fig. 3. In the regions influenced by Loop Current and Loop Current eddies, this figure shows high RMSE for SSH in the Free run (Fig. 3a) and large RMSE reductions in two data assimilative runs (Fig. 3b-c). In contrast, the reductions in SST RMSEs are more spatially homogeneous. A summary of the overall RMSE, the bias, and the correlation coefficient (Coef) for physical variables from the free run and two data assimilative runs are shown in Table 1. In general, the two data-assimilative runs both significantly improved SSH and SST with reduced RMSEs and increased correlation coefficients. Although the two data-assimilative runs tend to underestimate the satellite observations of SST, the bias (-0.06 °C) is relatively small.

The correction of mesoscale features by data assimilation was not limited to the surface but extend to the subsurface and even deep waters. Specifically, the two assimilative runs corrected the position, the amplitude, and the polarity of mesoscale eddies, and hence better represented the elevated and depressed thermoclines within these eddies (Fig. 4). The most noticeable improvement (by 60∼61%) was witnessed by the Float 287 which captured a newly detached Loop Current eddy with features of high SSH and depressed thermoclines during July and October. In addition, assimilation of

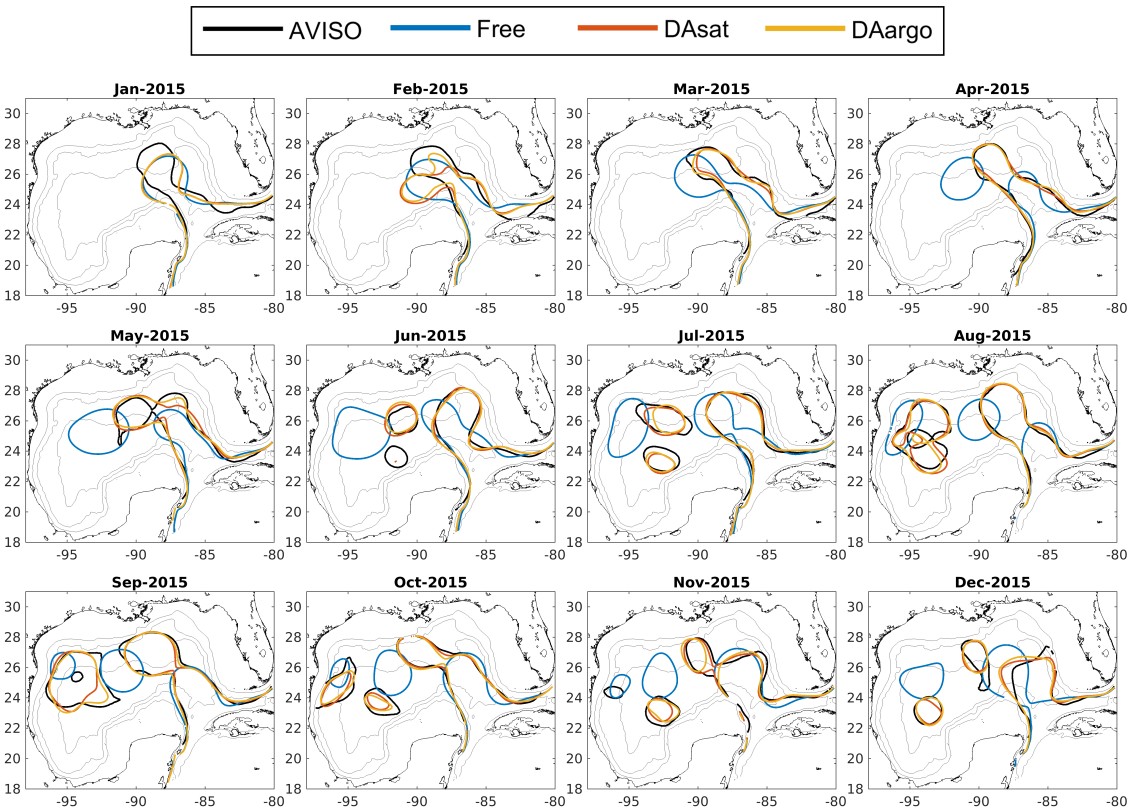

**Figure 2.** Monthly averaged Loop Current and Loop Current eddies based on 10-cm SSH contour from satellite data (black), free run (blue), DAsat run (orange), and DAargo run (yellow). The gray contours represent the isobath of 200 m, 1000 m, and 3000 m.

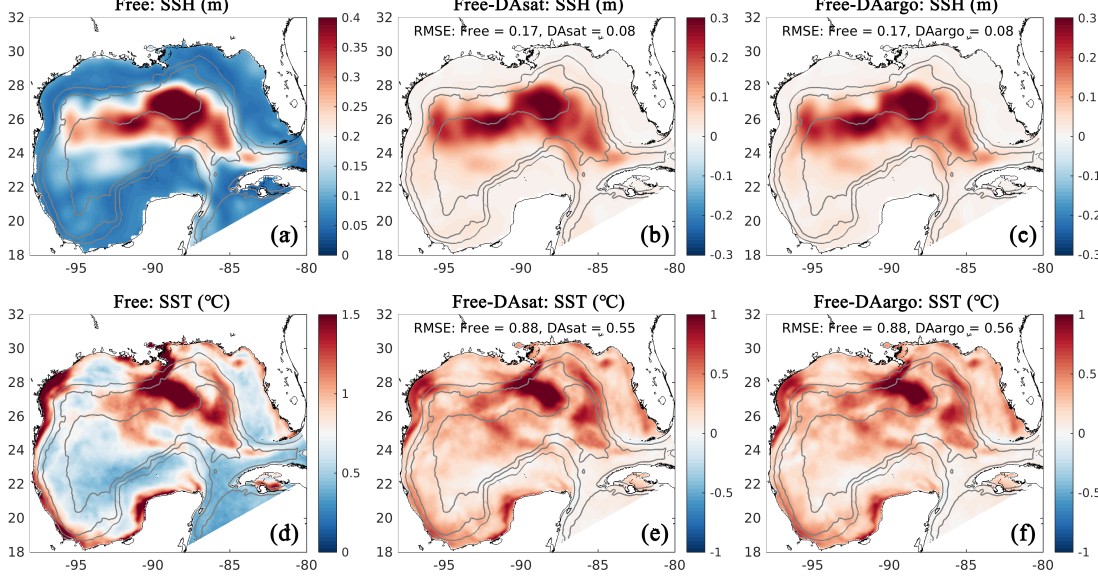

**Figure 3.** Spatial map of root-mean-square-error (RMSE) in the free run (a, d) and its differences between the free run and the two data-assimilative runs for SSH and SST (b, c, e, f). Positive values represent improvements while negative values represent deteriorations by data assimilation. Gray contours represent the 300-, 1000-, and 3000-m isobaths.

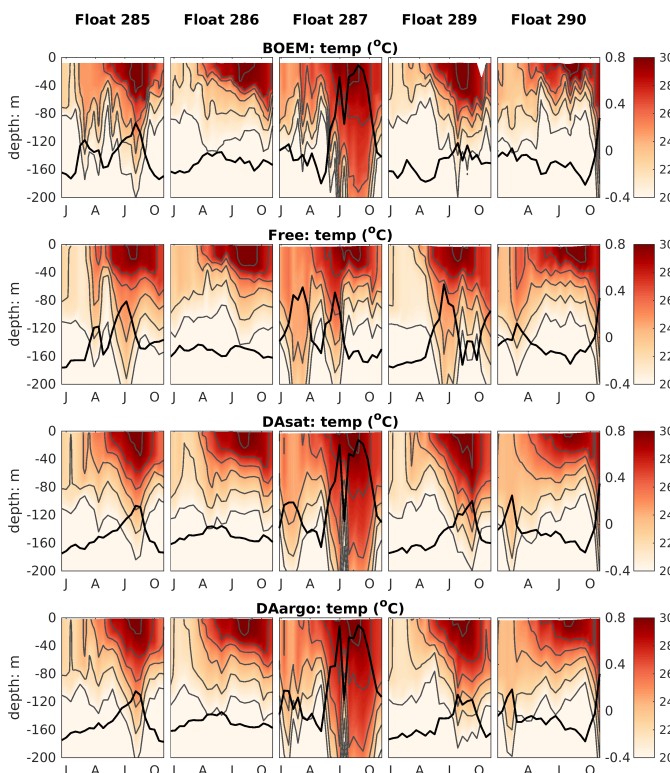

**Figure 4.** Vertical distributions of temperature from BOEM floats, the Free run, the DAsat run, and the DAargo run. Gray lines represent isothermal lines with an interval of $2°C$. Thick black lines represent SSH. The observed SSH is obtained from the matching record of altimeter observations.

Argo T-S profiles in the DAargo run led to slightly further improvements in the subsurface temperature distributions when compared with the DAsat run. For instance, although the DAsat run greatly improved subsurface temperature distributions along the trajectory of Float 285, an underestimation of temperature at about 200 m depth remains within the peak of the anticyclonic eddy. Corrections imposed by assimilating Argo profiles increased temperature here and decreased the bias from observations. These small but localized further improvements can also be observed by other floats, e.g. in July-October for Float 289 and February for Float 290.

In general, assimilating the satellite data in the DAsat run resulted in large reductions in RMSEs of 3D temperature (by 46%∼48%; Table 1) and salinity (by 36%∼39%; Table 1) with respect to Argo floats and BOEM floats (Fig. 5). The reductions extend to over 1,000 m and about 800 m depth for temperature and salinity, respectively. It should be noted again that data from both Argo and BOEM floats are independent in the DAsat run. Although assimilating the Argo profiles in the DAargo run only yields marginal further improvements in RMSEs of temperature (∼3%) and salinity (∼5%), it notably reduces the overestimation of temperature that occurs below the surface in the DAsat run (Table 1).

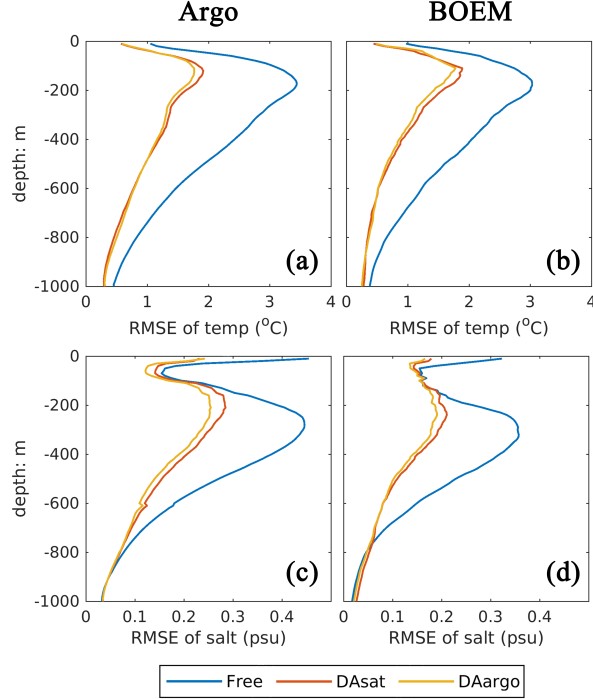

**Figure 5.** Vertical profiles of root-mean-square-error (RMSE) for temperature and salinity with respect to Argo and BOEM floats.

## 3.2 Assimilation impacts on biological properties

Assimilating satellite observations in the DAsat run reduced RMSEs of surface chlorophyll almost everywhere with only 3% of the model domain experiencing degradation (Fig. 6b). Although large reductions in RMSE were achieved in the coastal regions, e.g., in the northern Gulf of Mexico, on Campeche Bank, and in Campeche Bay, the simulated chlorophyll concentrations remained much lower than the satellite estimates because of high observational uncertainties and a large background misfit in the Free run (Fig. 6a). This was expected because the biological model was optimized for the open Gulf (Wang et al., 2020). Table 2 shows the RMSE, the bias, and the correlation coefficient for biological variables from the Free run and the data assimilative runs. A relative reduction in RMSE equal to or exceeding 10% is considered as a significant improvement. In the open Gulf, encompassed by the 1,000-m isobath, the overall RMSE of surface chlorophyll was reduced by 19% from 0.13 mg m$^{-3}$ in the Free run to 0.11 mg m$^{-3}$ in the DAsat run (Table 2). In addition, the correlation coefficient increased from 0.52 to 0.68. Assimilating Argo T-S profiles in the DAargo run led to lower reductions in the overall RMSEs of surface chlorophyll (Table 2) and even more deteriorations (Fig. 6c).

To evaluate the impacts of data assimilation on subsurface biological properties, the temporal evolution of nitrate in different model experiments is shown in Fig. 7 in compar-

**Table 1.** The root-mean-square-error (RMSE), Bias, and correlation coefficient (Corr) for SSH, SST, as well as vertical profiles of temperature and salinity from Argo and BOEM floats. Percentages in the parentheses represent the relative reductions in RMSE values. Since the spatial- and temporal- average of mismatch between the modelled and observed SSH is removed, the bias of SSH is not shown here.

| | SSH | SST | Argo | | Boem | |
|---|---|---|---|---|---|---|
| | **(m)** | **(°C)** | **Temp (°C)** | **salt** | **Temp (°C)** | **salt** |
| | | | **RMSE** | | | |
| **Free** | 0.17 | 0.88 | 1.70 | 0.22 | 1.55 | 0.18 |
| **DAsat** | 0.08 (54%) | 0.55 (37%) | 0.89 (48%) | 0.14 (36%) | 0.83 (46%) | 0.11 (39%) |
| **DAargo** | 0.08 (54%) | 0.56 (36%) | 0.86 (49%) | 0.13 (41%) | 0.79 (49%) | 0.10 (44%) |
| | | | **Bias** | | | |
| **Free** | – | 0.00 | 0.07 | 0.02 | 0.26 | 0.03 |
| **DAsat** | – | -0.06 | 0.12 | 0.02 | 0.24 | 0.02 |
| **DAargo** | – | -0.06 | 0.06 | 0.02 | 0.21 | 0.02 |
| | | | **Corr** | | | |
| **Free** | 0.72 | 0.96 | 0.96 | 0.92 | 0.97 | 0.95 |
| **DAsat** | 0.94 | 0.98 | 0.99 | 0.97 | 0.99 | 0.98 |
| **DAargo** | 0.94 | 0.98 | 0.99 | 0.97 | 0.99 | 0.98 |

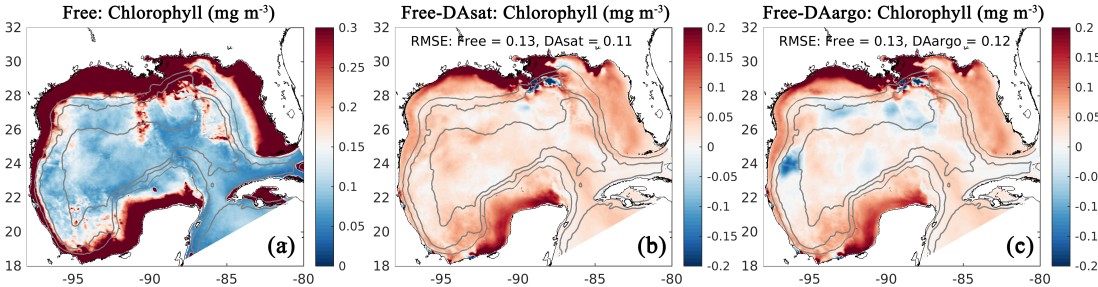

**Figure 6.** The same as Figure 3 except for surface chlorophyll.

ison to nitrate estimated based on its climatological relationship with temperature. The temperature-based nitrate tends to be overestimated in the upper layers (Fig. S1). Because of its high correlation with temperature, the nitrate distribution was modulated in the two assimilative runs along with the improvement in temperature fields. For instance, the two assimilative runs reproduce the Loop Current eddy observed by Float 287, and hence capture the depressed thermoclines that are not present in the Free run (Fig. 4). At the same time, the nitraclines are also depressed and the nitrate concentrations become lower within this Loop Current eddy (Fig. 7). As a result, the unbiased RMSE of nitrate following this float is reduced by 40% in the DAsat run and 38% in the DAargo run. These depressed (upwelled) nitraclines due to the increase (decrease) of SSH by data assimilation can also be observed elsewhere, e.g. in August of float 285, in April-July of the float 286, January-April of Float 287, and in August-October of Float 290, although the amplitude of these mesoscale eddies is smaller. In general, data assimilation improved the overall agreement of subsurface nitrate with correlation coefficients and decreased RMSEs by 28% and 30% in the DAsat and DAargo runs relative to the Free run (Table 2).

The impacts of assimilation on subsurface chlorophyll are more complicated because of the high nonlinearity of the model with regard to chlorophyll. Although the mean vertical profiles of chlorophyll are well reproduced in all three experiments (Fig. S2), all failed to resolve the high spatiotemporal variability in subsurface chlorophyll which is at least partly due to the presence of mesoscale eddies (Fig. 8). As a result, assimilation improved subsurface chlorophyll RMSEs marginally even in the Loop Current eddy of Float 287 where the most noticeable improvements of temperature (~60%) and nitrate (~40%) RMSEs were obtained. Results for phytoplankton and POC are similar as for chlorophyll although the reductions in their RMSEs are larger because assimilating the satellite data reduces their biases especially in the upper layer (Fig. S2, Table 2).

The model's inability to reproduce the spatiotemporal variability of subsurface chlorophyll is also reflected by the positions of the deep chlorophyll maximum (DCM, denoted by red lines in Fig. 8). As a ubiquitous phenomenon in the oligotrophic regions, a distinct DCM is observed throughout the whole year in the open Gulf of Mexico and its depth is inversely correlated with SSH (correlation coefficient = -0.6). Although the mean position and magnitude of the DCM are

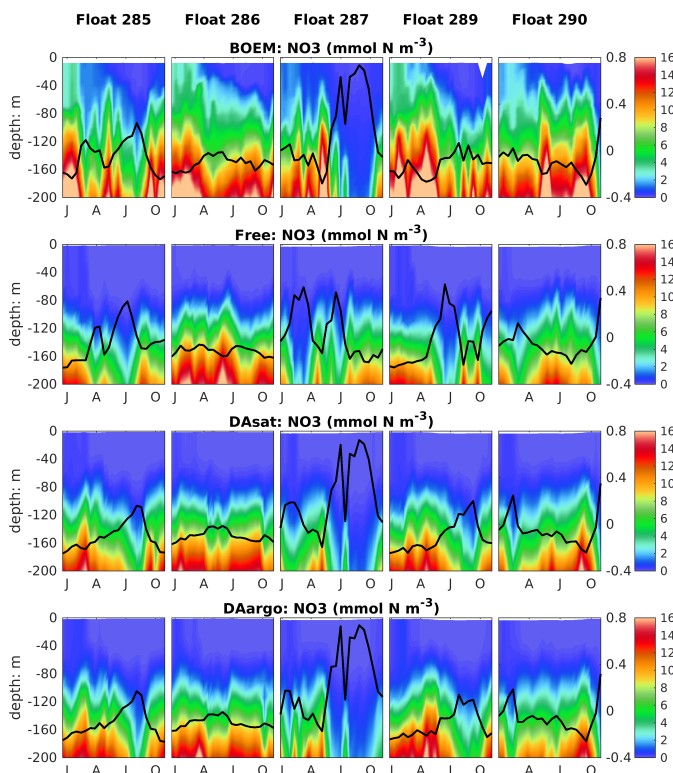

**Figure 7.** Vertical distributions of nitrate which are estimated based on its climatological relationship with temperature and modelled by different experiments, superimposed with the SSH (black thick lines).

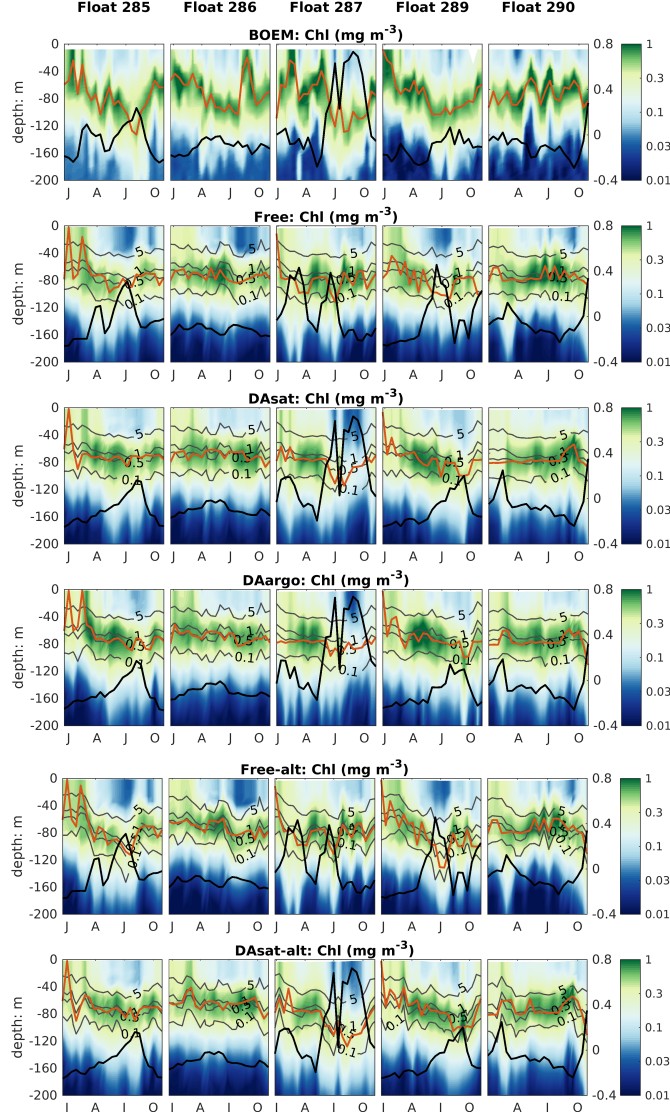

**Figure 8.** Same as Figure 4 but for chlorophyll. Gray contours represent the simulated isolumes and red lines represent the depth of deep chlorophyll maximum. Thick black lines represent SSH.

well reproduced by the model with and without data assimilation (Fig. S2), the simulated DCM depth is much more stable and less sensitive to SSH variations. As a result, the reduction in the RMSE of DCM depth is limited to 18% in DAsat run but is significant (Table 2).

### 3.3 Sensitivity of subsurface chlorophyll to the light attenuation parameterization

Both with and without data assimilation, the alternative parameterization led to higher correlations between simulated SSH and DCM depth with correlation coefficient of -0.60 in Free-alt run and -0.67 in DAsat-alt run. As a result, the alternative parameterization produces slightly lower RMSEs and a higher correlation coefficient for DCM depth (Table 2), and yields larger improvements in chlorophyll within the Loop Current eddy of Float 287 (Fig. 8). To illustrate the underlying reasons, the mean vertical profiles of nitrate, the intensity of photosynthetically active radiation (PAR), the chlorophyll, and the phytoplankton within the center of this Loop Current eddy are shown in Fig. 9. When using the original parameterization, assimilating the satellite data depresses the DCM depth from 70 m in the Free run to 90 m in the DAsat run, yet with a considerable bias of 20 m when compared

to the observations. However, the chlorophyll is underestimated in the DAsat run and as a result its RMSEs are barely improved. In contrast, in the DAsat-alt run the DCM depth is corrected to 120 m, in agreement with the observations, and represents the vertical chlorophyll distribution more accurately although the nitrate profile is almost the same as in DAsat run. This was because the alternative parameterization accounted for the elevated PAR intensity as a response to reduced chlorophyll concentrations in the upper layer, which in turn facilitated the synthesis of chlorophyll and hence corrected their concentrations toward the observations.

**Table 2.** The root-mean-square-error (RMSE), Bias, and correlation coefficient (Corr) for surface chlorophyll in the open gulf, vertical profiles of NO₃, chlorophyll, phytoplankton, and POC, as well as the depth of deep chlorophyll maximum with respect to observations from BOEM floats. Percentages in the parentheses represent the relative reductions in RMSE values. Only a reduction in RMSE larger than or equal to 10% is considered as a significant improvement. The NO₃ is estimated based on its climatological relationship with temperature. Since the estimated NO₃ tends to be overestimated in warm regions, the unbiased RMSE of NO₃ is reported and the bias is not shown here.

| | SChl (mg m$^{-3}$) | NO₃ (mmol N m$^{-3}$) | Chlorophyll (mmol N m$^{-3}$) | Phytoplankton (mmol N m$^{-3}$) | POC (mmol N m$^{-3}$) | DCM depth (m) |
|---|---|---|---|---|---|---|
| | | | **RMSE** | | | |
| **Free** | 0.13 | 3.71 | 0.18 | 0.11 | 18.62 | 25.48 |
| **DAsat** | 0.11 (19%) | 2.66 (28%) | 0.17 (6%) | 0.10 (9%) | 16.46 (12%) | 21.08 (18%) |
| **DAargo** | 0.12 (9%) | 2.58 (30%) | 0.17 (6%) | 0.10 (9%) | 16.77 (10%) | 22.39 (12%) |
| **Free_alt** | 0.17 | 3.71 | 0.18 | 0.11 | 17.55 | 24.35 |
| **DAsat_alt** | 0.13 (26%) | 2.63 (29%) | 0.17 (6%) | 0.10 (9%) | 15.53 (12%) | 20.42 (16%) |
| | | | **Bias** | | | |
| **Free** | -0.01 | – | -0.04 | -0.02 | -8.01 | -0.98 |
| **DAsat** | 0.02 | – | -0.04 | -0.01 | -5.05 | 0.45 |
| **DAargo** | 0.03 | – | -0.02 | -0.01 | -3.84 | 2.59 |
| **Free_alt** | 0.00 | – | -0.04 | -0.02 | -6.57 | -1.09 |
| **DAsat_alt** | 0.03 | – | -0.03 | -0.00 | -3.15 | 1.83 |
| | | | **Corr** | | | |
| **Free** | 0.52 | 0.94 | 0.73 | 0.72 | 0.63 | 0.25 |
| **DAsat** | 0.68 | 0.97 | 0.76 | 0.75 | 0.71 | 0.50 |
| **DAargo** | 0.65 | 0.97 | 0.74 | 0.75 | 0.70 | 0.45 |
| **Free_alt** | 0.58 | 0.94 | 0.73 | 0.72 | 0.64 | 0.43 |
| **DAsat_alt** | 0.70 | 0.97 | 0.76 | 0.75 | 0.72 | 0.58 |

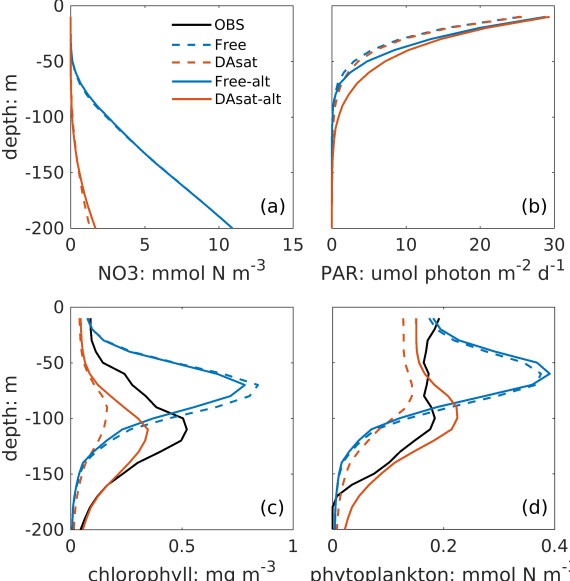

**Figure 9.** Mean vertical profiles of nitrate, light intensity (photosynthetically active radiation, PAR), chlorophyll, and phytoplankton within the center of the newly detached Loop Current eddy from the Free run, the DAsat run, the Free-alt run, and the DAsat-alt run.

## 4 Discussion

We implemented a coupled data assimilation scheme for jointly assimilating physical and biological observations in a biogeochemical model and evaluated to what degree satellite observations can inform subsurface distributions, especially of biological properties. The degree to which the data assimilation impact can depend on model calibration was tested by using an alternative light parametrization. Although biological data assimilation has received much attention in recent years, observations that are assimilated and used in skill assessment are typically limited to the surface ocean. The increasing availability of BGC-Argo data now makes it possible to validate and improve model performance below the surface (Cossarini et al., 2019; Salon et al., 2019; Terzić et al., 2019; Wang et al., 2020) but so far these observations are too sparse for sequential assimilation in three dimensions; hence, relevant applications are limited to idealized twin experiments (Ford, 2021; Yu et al., 2018) and a few specific regions with high float densities, e.g. the Mediterranean Sea (Cossarini et al., 2019). In addition, since a biogeochemical model is coupled to a physical model, assimilating physical observations theoretically should confer improvements in the biological model through correcting the circulation (e.g. Fiechter et al., 2011; Raghukumar et al., 2015; Song et al., 2016a, b) and potentially by providing additional constraints via multivariate updates to biological variables (e.g. Goodliff et al., 2019; Yu et al., 2018). This is particularly important when the physical model is biased (Yu et al., 2018).

Our study shows that assimilating satellite data (DAsat run) can constrain the main circulation features in the Gulf of Mexico, i.e. the Loop Current and its associated mesoscale

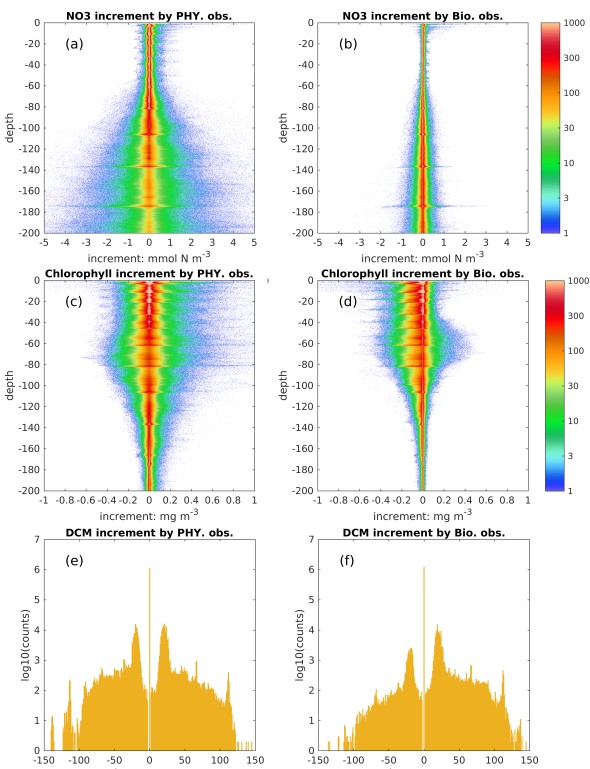

**Figure 10.** Histogram of increment in nitrate (mmol N m$^{-3}$), chlorophyll (mg m$^{-3}$)), and DCM depth (m) obtained by assimilating physical and biological observations.

eddies. Temperature and salinity are also improved down to ∼1,000 m depth because of the correction of mesoscale eddies. When calculating the reductions in RMSE for SSH and each single profile of temperature and salinity, we find that the improvement in SSH is highly correlated with those in temperature (correlation coefficient = 0.96) and salinity (correlation coefficient = 0.92, Fig. S3). Assimilating the satellite data also improves subsurface nitrate because it is tightly correlated with the density structure expressed by SSH and temperature profiles. However, improvements in temperature and nitrate do not necessarily yield better simulations of chlorophyll or phytoplankton because they tend to be light limited below the surface. In our biogeochemical model, the light intensity is attenuated by water and chlorophyll, and is not directly updated by the data assimilation scheme but only adjusted indirectly through changes in chlorophyll during forecast steps. This, in turn, impacts the synthesis of chlorophyll and growth of phytoplankton. However, in the original parameterization, light attenuation is mainly controlled by water depth and much less sensitive to chlorophyll concentrations than it appears to be in reality. By applying an alternative light parameterization with more pronounced self-shading by chlorophyll, the subsurface chlorophyll and phytoplankton distributions are further improved after assimilating the satellite data. These results show that the biological variables can be improved through model dynamical response to data assimilation. However, the efficiency of this mechanism depends on the accuracy of the biological model. That's why data assimilation generally benefits from a well-calibrated model. For example, the usage of suboptimal biological parameters can yield a substantial degradation of data assimilation efficiency, especially with respect to unobserved variables (Song et al., 2016a). Although BGC-Argo profiles so far are insufficient for sequential assimilation, they can provide substantial benefits to the biogeochemical prediction by enabling *a priori* model tuning, e.g. of biological parameter values (Wang et al., 2020) and the key parameterization schemes (Terzić et al., 2019).

In addition to the model's dynamical response, the biological fields can be directly updated by physical and biological observations through multivariate covariances. To distinguish their influence, we show the increments obtained from assimilating each observation type in the DAsat run (Fig. 10). The increment of DCM depth is defined analogously to other state variables as changes due to the update. As shown in Fig. 10a, b, assimilating physical observations has a much stronger impact than biological observations on nitrate and therefore we conclude that the improvement of nitrate in this study is mainly obtained from assimilating physical observations. This is consistent with previous studies (e.g. Ciavatta et al., 2018; Skákala et al., 2018; Teruzzi et al., 2018) where assimilating surface chlorophyll had little impact on nitrate and even degraded it in both variational and sequential data assimilation. In variational data assimilation, it is hard to define the background errors accurately (Mattern et al., 2017; Teruzzi et al., 2018) and the biological model can fit itself to observed chlorophyll by many different pathways, e.g. direct changes of biomass or an indirect way through nitrate. However, observations are often insufficient to provide this information (Mattern et al., 2017). In sequential data assimilation, the multivariate covariance between surface chlorophyll and subsurface nitrate can be considered but typically this covariance is not linear or constant. For instance, Fontana et al. (2013) assimilated satellite surface chlorophyll into a biological model in the North Atlantic and found that subsurface nitrate was barely influenced because it was weakly correlated with surface chlorophyll, leading the authors to suggested that it is impossible to fully constrain a 3D biogeochemical model by only assimilating the surface chlorophyll. This issue remains when assimilating the surface chlorophyll to update other biological variables (Yu et al., 2018), e.g. phytoplankton functional groups (Ciavatta et al., 2018).

In contrast to nitrate, assimilating satellite data of physical and biological observations have a comparable influence on subsurface chlorophyll (Fig. 10c-f). Specifically, they can change subsurface chlorophyll concentrations even below 100 m depth and vertical structures of chlorophyll by adjusting the DCM depth, e.g., there are 10% and 5% of profiles with changes in DCM depth exceeding ±20 m due to

the update of physical and biological observations, respectively. Because currently BGC-Argo profiles are sparse, i.e. only 14 profiles are available at all update steps, it is hard to draw definitive conclusions about these impacts on chlorophyll and DCM depth.

Assimilating Argo T-S profiles in the DAargo run yields slightly further improvements with respect to independent profiles of temperature and salinity, similar to the twin experiments in Yu et al. (2019). To diagnose it, we calculate the root-mean-square-difference (RMSD) of temperature between two data assimilative runs with respect to each profile from the BOEM floats. In general, the RMSD between two data assimilative runs decreases with distance to the nearest Argo profiles that have been assimilated recently but shows no significant decreasing trends with the days after update (Fig. S4). This means that the differences induced by assimilating Argo profiles are sustained locally by model dynamical adjustments. The overall similarities between two data assimilative runs (i.e. DAsat and DAargo runs) in Fig. 4 can be explained to some extent by the large distances between BOEM and Argo profiles. However, it doesn't mean that increasing the localization radius can necessarily improve the data assimilation performance. We note that the current localization radius was determined in Yu et al. (2019). The additional benefits in physical properties obtained by assimilating Argo T&S profiles are also translated into the simulation of subsurface nitrate but not into other biological fields, i.e. chlorophyll, phytoplankton, and POC. Moreover, assimilating the Argo T-S profiles can even degrade surface chlorophyll because of spurious correlations. This issue has been also reported in a recent study (Goodliff et al., 2019) which assimilated sea surface temperature to update both physical and biological variables and this issue was alleviated by muting the multivariate update of phytoplankton, zooplankton, and detritus.

In general, coupled data assimilation of both physical and biological satellite observations can improve subsurface biological properties because it benefits from the high correlations of some biological distributions, especially nutrients, with the vertical density structure and because of the dynamical responses to improvements in circulation in the forecast step. However, this is preconditioned on the coupled model being well calibrated *a priori*. Therefore, this study provides an intermediate step toward 3D updates of biological properties before the BGC-Argo profiles will ultimately become more abundant.

## 5    Conclusions

In this study, a coupled data assimilation scheme for both physical and biological satellite observations was implemented to investigate whether these observations can inform subsurface distributions. In addition, Argo T-S profiles were assimilated to assess their impact beyond satellite observations. The multivariate update was applied by using the covariance structure between physical and biological variables. The Gulf of Mexico was selected as the study region because the dominant physical features, the Loop Current and its associated mesoscale eddies, are stochastic and can influence the biological properties in three dimensions substantially. Our results show that assimilating satellite data leads to significant improvements in the simulation of SSH and SST, and also projects these improvements from the surface to about 1,000 m depth for temperature and salinity as shown by an assessment of the independent BGC-Argo profiles. With respect to biological fields, the subsurface nitrate distribution benefits greatly from the tight correlation with density and the improved fidelity of mesoscale features. However, initially there were only slight improvements in other biological variables below the surface, i.e. chlorophyll, phytoplankton, and POC, because a suboptimal light parameterization did not react to the changed chlorophyll concentrations appropriately and failed to provide accurate feedbacks on the synthesis of chlorophyll and growth of phytoplankton. We tested an alternative light parameterization with a larger relative contribution from chlorophyll to light attenuation. As a result, the subsurface chlorophyll and phytoplankton were further improved. This highlights the importance of *a priori* tuning to achieve better assimilation performance. Finally, assimilating the Argo T-S profiles on top of satellite observations yields slight further improvements with respect to independent vertical profiles of temperature and salinity, which also translated into improvements in subsurface nitrate.

*Code and data availability.* The ROMS model code can be accessed at http://www.myroms.com (last access: 16 June 2016). HYCOM data can be downloaded at http://tds.hycom.org/thredds/dodsC/datasets (last access: 16 August 2018). Profiling data from the BGC-Argo floats are available at the National Oceanographic Data Center (NOAA), https://data.nodc.noaa.gov/cgi-bin/iso?id=gov.noaa.nodc:159562 (Hamilton and Leidos, 2017)

*Author contributions.* BW and KF conceived the study. BW carried out data assimilation experiments and analyses. LY provided data assimilation techniques. BW and KF discussed the results and wrote the paper with contributions from LY.

*Competing interests.* The authors declare that they have no conflict of interest.

*Acknowledgements.* This research was funded by the Gulf of Mexico Research Initiative (GoMRI-V-487).

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
