# Peer review of "Can assimilation of satellite observations improve subsurface biological properties in a numerical model? A case study for the Gulf of Mexico"

_Ocean Science, 2021_

## Author Comment (AC1)

**Responses to Reviewer 1**

Below the complete reviewer comments are shown in black font along with detailed responses to each comment in blue font.

Review:
This is a very interesting and valuable contribution to the present scientific literature addressing the impact of (DEnKF) assimilation of physical and biogeochemical data, including Argo floats, on the simulated state of the ocean. More specifically, the paper addresses an important question of the impact of (mostly) surface data assimilation on sub-surface physics and biogeochemistry, focusing on the Gulf of Mexico. The simulated physical and biogeochemical tracers are validated with an independent BGC-Argo data-set. The paper is well written and I have mostly only minor comments to address, which can be found below. One remark: please note that in the downloaded .pdf version of the manuscript the first digits of the three-digit line numbers are running off the left side of the page, so beneath the section 1 I avoid referring to the line numbers and refer to the paragraph and section number instead.
Response: We thank the reviewer for the constructive comments and suggestions which will be very helpful as we revise the manuscript.

Section 1:
The introductory section is nicely written and the only comments that I have are regarding the use of references:
- in particular on line 52 when the paper talks about assimilation of optical properties, the included references are very far from exhaustive. On top of my head I could add to the list the following papers: 1. Jones et al (2016), Biogeosciences, https://doi.org/10.5194/bg-13-6441-2016, 2. Gregg and Rousseaux (2017), Frontiers in Marine Science, https://doi.org/10.3389/fmars.2017.00060, 3. Skakala et al (2020), Journal of Geophysical Research: Oceans, https://doi.org/10.1029/2020JC016122. I completely understand that sometimes there simply are too many references and you cite only the first ``pioneering'' papers, but in such case it would be good to put "e.g." in front of the citations, or add after the citations "with few more recent follow-ups", or something similar. Otherwise this might look like the cited papers are all that has been published on the topic and it might lead to the other papers being omitted when someone new-to-the-area uses your references as part of literature review on the topic. Similarly to this, I would make sure you somehow emphasize that the citations aren't exhaustive also in the other cases, e.g. the phytoplankton size-class chlorophyll has been assimilated also in Ciavatta et al 2019, Journal of Geophysical Research: Oceans, https://doi.org/10.1029/2019JC015128 and for total surface chlorophyll there's many references left out.
Response: Thank you very much for pointing out this. We will add these references and revise the corresponding text in our revised manuscript as suggested.

- more importantly the text on the lines 65-70 suggests that the subsurface validation was done only in the 2 cited studies and only on climatological, or basin-wide scale. Firstly, I believe it is appropriate to include here the Cossarini et al 2019 reference that is cited elsewhere in the paper (which looks at vertical profiles along BGC-Argo locations), but secondly there is a new study of Skakala et al (2021), Journal of Geophysical Research: Oceans, https://doi.org/10.1029/2020JC016649 that validates both free runs and surface data assimilative runs along glider trajectories (in the North-West European Shelf) to determine the impact of surface data assimilation (and some other DA as well) on the simulated sub-surface tracers. This study, although it uses a very different, 3DVar, system, is of direct relevance here, since it addresses similar questions to this submitted manuscript, and, as mentioned before, it does model validation along a specific glider 3D transect (which is of course spatially limiting), rather than on basin-wide scales. Btw. please note that there is another study that assimilated BGC-Argo oxygen data that perhaps deserves to be cited in this paper: Verdy and Mazloff, Journal of Geophysical Research: Oceans, https://doi.org/10.1002/2016JC012650.
Response: We will include these references in our revised manuscript and rephrase this paragraph into:

*"The insufficient availability of subsurface and interior ocean biogeochemical observations is not only reflected in the immaturity of biogeochemical data assimilation but also its skill assessment. When compared with the surface, the subsurface has received less attention in skill assessments of biogeochemical data assimilation systems. Although there already have been studies which compared their vertical structures with in-situ observations and/or climatological datasets (e.g. Fontana et al., 2013; Ford and Barciela, 2017; Mattern et al., 2017; Ourmières et al., 2009; Teruzzi et al., 2014), these validations are often limited to low spatio-temporal resolution. The recent growth of autonomous observation systems, esp. BGC-Argo floats and gliders, make it possible to evaluate biogeochemical data assimilation systems below the surface in high resolution (e.g. Cossarini et al., 2019; Salon et al., 2019; Skákala et al., 2021; Verdy and Mazloff, 2017)."*

Section 2:
I find it would be perhaps beneficial for the reader to provide slightly more information on the system set-up and maybe a bit rearrange the text. In particular:
- could you please mark the exact model domain? The Fig.1 shows the Gulf of Mexico region with its bathymetry, but I guess what's in Fig.1 is not precisely the spatial model domain (?) Or is it the red rectangle in Fig.1, which however isn't explained (?) Maybe you can mask out the ``irrelevant'' regions?
Response: The model domain is represented by the red rectangle. We will make it clearer in the main text and figure caption of our revised manuscript.

- a very minor (bottom of first paragraph in the section 2.1): could you please provide a rough km scale for the 1/8 degree resolution. Of course this is slightly variable depending on the latitude, but is it something like ~12km?
Response: The longitudinal distance is about 12 km and the latitudinal distance is about 14 km. We will add this into our revised manuscript as suggested.

- again extremely minor: just beneath Eq.2 "measurement operator" should be "observation operator"?
Response: We will revise it as suggested.

- I would perhaps suggest to put the first two paragraphs in section 2.4 behind the section 2.1, or maybe behind the section 2.2, so you say clearly how the ensemble is generated, how many members are run (and so on), around the same time when you talk about ensemble DA. I got firstly confused how little information is provided about the ensemble generation until I realized it's been put behind the "Observations" section.
Response: Thank you very much for this suggestion. We will move these two paragraphs to behind the section 2.2 in our revised manuscript.

- regarding the ensemble generation: how was the parameter sensitivity determined, in order to perturb the ``right'' parameters to account for the model uncertainty? Physical model parameters weren't perturbed (e.g. vertical diffusion parameter values)? If so, why? I know 20 member ensemble is itself limiting, but perhaps some more reasoning on how the ensemble was generated would be desirable.
Response:
The biological parameters were selected by sensitivity tests. Specifically, we ran a 1D version of this model multiple times by incrementally perturbing one parameter at a time and setting other parameters unchanged (hereafter as **Test case**). Each parameter ($p$) was perturbed by multiplying factors ranging from 0.25 to 1.75 with an increment of 0.25. Then parameters were sorted based on their sensitivity which was quantified by the normalized absolute differences from the unperturbed run (hereafter as **Base case**):

$$Q(y,p) = \frac{1}{m} \sum_{i=1}^{m} \frac{1}{n} \sum_{j=1}^{n} \frac{|y_{Base} - y_{Test}|}{y_{Base}}$$

where $m$ is the number of parameter increments (here 7) and $n$ is the number of base-test pairs including all 1D model grid cells throughout the whole simulation period. Based on the sensitivity tests, the most

sensitive parameters were selected and perturbed in our data assimilation experiments. The details about the 1D model can be found in the section 3.3 of Wang et al. (2020). In our revised manuscript, we will add explanation about the choice of sensitive biological parameters.

The physical model parameters (e.g. vertical diffusion coefficient) weren't perturbed in this study. To create an ensemble with sufficient spread, we perturbed the model initial conditions, the open boundary conditions, and the wind forcing. Details in these perturbations can be seen in the section 2.4 of our original manuscript.

It is worth noting that this data assimilation framework and configurations (e.g. observational errors, ensemble perturbations, ensemble members, and etc.) are obtained from Yu et al. (2019) which performed twin experiments in the same region (the Gulf of Mexico). In this study, we extend the work of Yu et al. (2019) to jointly assimilate the physical and biological observations.

- it seems that the observation uncertainties don't come with the observation product (?), but based on what you estimate the 35% observational uncertainty for total chlorophyll, or the physical observation errors? This might be of importance, since you are aiming at estimating model variances quite accurately with the ensemble, so the true state estimate can be quite sensitive to the observation error estimates.
Response: An observational error of 35% (or 30%) is a common practice to specify the error of satellite surface chlorophyll (e.g. Fontana et al., 2013; Ford, 2021; Ford and Barciela, 2017; Hu et al., 2012; Mattern et al., 2017; Santana-Falcón et al., 2020; Song et al., 2016; Yu et al., 2018). The observational errors of SSH and SST are also based on references (Song et al., 2016; Yu et al., 2018, 2019).

- btw do you assimilate chlorophyll, or log-chlorophyll? If the former, how do you treat the problem of non-Gaussianity?
Response: We assimilate the actual chlorophyll concentrations in this study. Based on our prior tests, assimilating log-chlorophyll can yield more improvements of the surface chlorophyll in coastal regions but degrade it in the deep ocean (with depth>1,000m). In addition, assimilating log-chlorophyll results in less improvements of subsurface biological properties, including the chlorophyll, phytoplankton, and POC. As this study focused on the subsurface biological properties in the deep ocean of the Gulf of Mexico, we decided to assimilate the actual chlorophyll. Also, there already have been successful examples which assimilated the actual chlorophyll (e.g. Hu et al., 2012; Yu et al., 2018). Nevertheless, we acknowledge that assimilating the actual chlorophyll is suboptimal because of its non-Gaussian nature and will state this more clearly in the revised manuscript.

–what exactly is the 7 day window assimilation at the end of third paragraph of section 2.3, and how exactly does it help with data sparsity, should be explained more in detail.. E.g. is this that you assimilate more data that have been otherwise merged in the OC CCI products?
Response: In this study, we applied update every week. Since the spatial coverage of surface chlorophyll is limited (with average of ~ 9.5%), we not only assimilate the daily records of surface chlorophyll at the date of update (e.g. 7 Jan 2015, the first update date in our data assimilative experiments), but also assimilate all daily records within 7 days before (e.g. from 1 Jan 2015 to 7 Jan 2015). Literally, the abundance of surface chlorophyll data that we assimilated will increase by approximately 7 times. We will explain this in more detail in our revised manuscript.

- I am not particularly great expert on EnKF, but maybe you could say for a general readership a bit more about how exactly are the correlation length-scales (especially the vertical) calculated from the ensemble? There is something on the spurious correlations and localisation, but perhaps you can say a bit more? Also I understand correctly that SSH, T & S are updated without updating also the horizontal velocities? Could that be also discussed slightly more?
Response:
The data assimilation framework and configurations (e.g. localization radius, observational errors, ensemble perturbations, ensemble members, and etc.) used in this study are the same as in Yu et al. (2019)

which has successfully performed twin experiments in the same region (the Gulf of Mexico). The localization radius was tuned by initial sensitivity tests in Yu et al. (2019). Specifically, the different values of localization radius (i.e. 50, 75, and 100km) were tested and the optimal value (50km) was selected. The vertical correlation length-scale is not used in this study.

In our data assimilation experiments, the 3D distributions of temperature and salinity were updated. The SSH and horizontal velocities were adjusted by dynamics in the following forecast steps. In the deep Gulf of Mexico, the circulation features (e.g. the Loop Current and its associated mesoscale eddies) are primarily geotropically balanced. Therefore, an update of temperature and salinity can improve the 3D circulation features in large scales effectively. This can be also supported by the twin experiments in Yu et al. (2019). As this study mainly focuses on the biological aspect, discussing the update of horizontal velocities may be a little distracting, but we will refer to the explanations in Yu et al. (2019) more clearly in the revised manuscript.

- the skill metrics at the end of section 2: why don't you also include the overall bias? I find that calculating bias always carries worthy information (at least worthy enough to be mentioned somewhere in the paper).
Response: We agree that the bias can carry supplementary information to the RMSE. As also suggested by the review 3, we will report the bias and correlation coefficient in our revised manuscript.

Section 3
Nice, just some relatively minor clarifications would help:
- section 3.2: I find it surprising that addition of Argo (physical) data to the assimilation (Daargo) has such impact on surface chl (Fig.6), especially if it didn't change much T & S (Fig.4)? Could you comment a bit more?
Response: The degradation of surface chlorophyll in DAargo run is caused by the multivariate update between the Argo T&S profiles and chlorophyll, rather than the dynamical adjustment. Figure r1 shows the forecast and analysis of surface chlorophyll at the first update date (7-Jan-2015). The white circles in Figure r1c represent the assimilated Argo T&S profiles. We can see that assimilating the Argo T&S profiles can yield extremely high surface chlorophyll around some profiles (e.g. the profile located at about 96.5°W, 24°N). This is due to the spurious correlations between the T&S profiles and surface chlorophyll. This issue has been also reported in Goodliff et al. (2019) which assimilates SST to update both physical and biological variables. In their study, the multivariate update can yield unrealistic chlorophyll concentrations no matter when the chlorophyll has been log-transformed. Muting the multivariate update on phytoplankton, zooplankton, and detritus can make chlorophyll concentrations more realistic.

[Figure]

Figure r1. The forecast and analysis of surface chlorophyll at the first data assimilation cycle (7-Jan-2015). The white circles in panel c represent the assimilated Argo T&S profiles.

Section 3.2 paragraph 2: where are the nitrate data coming from? Maybe I'm getting confused, but nitrate data weren't mentioned between the BGC Argo data in section 2.
Response: The NO3 is not measured by the BGC-Argo floats. It is estimated based on its empirical relationship with temperature (Figure S1 of our original manuscript), which is derived from the

climatological data in the Gulf of Mexico from the World Ocean Atlas (WOA). The description of estimated NO3 is in the end of the section 2.4 in our original manuscript. To make it clearer, we will move it to the end of section 2.3 (observations) in our revised manuscript.

specific Figure-related comments:
Fig.3: although I understand what the Figure is trying to demonstrate, it might be better to show the model –observation differences for the different runs, or at least show them for the free run next to the existing panels?
Response: Figure 3 in our original manuscript aimed to show reductions in RMSE and therefore the improvement obtained by data assimilation. We agree with the reviewer and we will add panels to show RMSE of the free run as a benchmark in our revised manuscript as suggested.

Fig.4: its interesting how little difference there is in subsurface T between sat DA and argo DA! Can you discuss how much of this is due to the assimilation length-scales and how much due to the model dynamical adjustment? Btw what was the relationship between the spatial positions of Argo floats and BOEM floats? This hasn't been shown..
Response:
    Figure r2 shows positions of Argo profiles (gray dots) at each data assimilation cycle (e.g. 7 Jan 2015, the first update date in our data assimilative experiments) and BOEM profiles (colored squares) before the next one (e.g. from 7 Jan 2015 to 14 Jan 2015). The solid black circles represent areas within one localization radius (50km) from each Argo profile. Colors of squares represent the days of each BOEM profile after each data assimilation cycle. As shown in Figure r2, most of BOEM profiles are outside of one localization radius from the Argo profiles and therefore are barely updated by assimilating the Argo profiles. Figure r3 shows the root-mean-square-difference (RMSD) of temperature from each BOEM profile between two data assimilative runs ($RMSD = \sqrt{\frac{1}{n}\sum(DA_{Sat} - DA_{Argo})^2}$). The x-axis represents days of each BOEM profile after each data assimilation cycle and the y-axis represents distance to the nearest Argo profile. In general, the RMSD between two data assimilative runs decreases with the distance but shows no significant decreasing trends with the days after update. This means that the differences induced by assimilating Argo profiles can be well sustained locally by model dynamical adjustments. The overall similarities between two data assimilative runs in Figure 4 can be explained to some extent by the large distances between BOEM and Argo profiles. However, it doesn't mean that increasing the localization radius can improve the data assimilation performance. We note that the current localization radius is determined by initial tests in Yu et al. (2019)

[Figure]

(continued)

[Figure]

Figure r2 Positions of Argo profiles (gray dots) at each data assimilation cycle and BOEM profiles (colored square) before the next one. Solid black circles represent areas within one localization radius from each Argo profile. Colors of squares represent the days after each data assimilation cycle.

[Figure]

Figure r3 The root-mean-square-difference (RMSD) of temperature from each BOEM profile between two data assimilative runs, DAsat and DAargo (indicated by the color). The x-axis represents days of each BOEM profile after each data assimilation cycle and the y-axis represents distance to the nearest Argo profile

Fig.7: perhaps in this case there is no need to reproduce exactly Fig.4, since the nitrate concentrations are very similar between the three simulations? Maybe it's better to have the first row and then differences between the DA and the free run, since this would show more clearly the changes introduced by DA? Also why there aren't the observed nitrate concentrations similarly to T in Fig.4? It's interesting that nitrate and surface chlorophyll aren't correlated as much as nitrate with temperature (Fig.10), I understand that's because of non-linearity, but in my region and model of experience this non-linearity still produces overall strong correlations, just highly variable in time (including changing signature)..

Response:

The NO3 distributions are quite different before and after data assimilation. They look similar in Figure 7 of our original manuscript possibly because the NO3 are plotted in log-scales. As suggested, we replotted Figure 7 (referred as Figure r4 here) to show NO3 distributions in the free run and increment of NO3 due to data assimilation in DAsat and DAargo runs. Red colors represent increases while blue colors represent decreases of NO3 by data assimilation. As shown in Figure r4, the NO3 distributions are significantly modified and improved by data assimilation. For example, the anticyclonic eddies which are not reproduced by the free run will depress the nitraclines and decrease the NO3 concentrations (e.g. during June of the float 286, during July and October of the float 287, and during August to October of the float 289) in the two data assimilative runs. In contrast, the data assimilation will increase NO3 concentrations when the SSH is overestimated in the free run and corrected in the two data assimilative runs (e.g. during April of Float 285, during February and April of Float 287, during April to July of Float 289).

We didn't show NO3 observations because NO3 is not measured by the BGC-Argo floats in the Gulf of Mexico. In this study, we estimated NO3 based on its climatological relationship with temperature (Figure S1 of our original manuscript which is replot as Figure r5 here) and compared it with the model results. The estimated NO3 distributions following the BGC-Argo floats are shown in Figure r4. However, the estimated NO3 tend to be overestimated in the high temperature regions (Figure r5), which typically

occurs within the euphotic layer. Therefore we used the unbiased root-mean-square-error to quantify the model-data misfit of NO3.

For the relatively weak correlations between surface NO3 and chlorophyll, the deep ocean of the Gulf of Mexico (depth>1,000m) is an oligotrophic region. Apart from NO3, ammonia (NH4) can be also used to support a large fraction of primary production. In addition, the photo-acclimation is accounted for in our biological model. The chlorophyll to phytoplankton carbon ratio can vary a lot due to environmental factors, i.e. light, temperature, and nutrient concentrations (Geider et al., 1997). All these features will increase nonlinearities of the biological model and make the correlations between surface NO3 and chlorophyll weak.

[Figure]

Figure r4 Vertical distributions of NO3 estimated based on its climatological relationship with temperature and modelled by the Free run, and increment of NO3 due to data assimilation in the DAsat and DAargo run. Black lines represent SSH.

[Figure]

Figure r5 Empirical relationship of temperature-NO3 derived from the World Ocean Atlas in the Gulf of Mexico. Colors indicate the number of observations within each bin.

**Reference**:

[revised manuscript text omitted]

---

## Author Comment (AC2)

**Responses to Reviewer 2**

Below the complete reviewer comments are shown in black font along with detailed responses to each comment in blue font.

**Review:**

The manuscript by Wang and Coauthors investigates the impact of the assimilatio of satellite surface observations to both physical and biogeochemical variables in a coupled model of the Gulf of Mexico physical and biogeochemical dynamics. Independent (nonassimilated) profiles form five BGC-Argo floats are used for validation. Results provided in the manuscript highlight interesting aspects on the capability of DA (data assimilation) to effectively correct ocean simulations, on the difficulties arising in coupled physical and biogeochemical assimilation, and on the relevant role played by prior model calibration in DA. The manuscript is well written and of interest for the scientific community considering recent and foreseen upgrades in physical-biogeochemical ocean DA. My review is limited to few points (most of them minor) that I think will further improve the manuscript quality.

Since line numbers are corrupted in the manuscript file, in the comments hereafter they are indicated by # followed by the part of line number visible in the manuscript. Page numbers are also provided together with line numbers.

Response: We thank the reviewer for the constructive comments and suggestions which will be very helpful as we revise the manuscript.

I suggest to introduce the alternative parametrization of the light absorption in a different way. In the manuscript it is currently described as an alternative that has been considered after investigating the results of previous simulations. I think that presenting this formulation as an alternative since the beginning would better emphasize the role of prior model calibration. Thus, I suggest to describe the alternative formulation not in a temporal framework (i.e., without specifying that it has been applied after previous simulations) as it is currently done in the abstract and in the manuscript sections. In particular: i) the first paragraph of Section 3.3 could be moved and adapted to Section 2.4; ii) in Section 2.4 the Authors could indicate that five (instead of three) simulations were performed; iii) it can be further stressed through the manuscript that the alternative formulation for the light absorption was adopted to investigate the sensitivity of subsurface DA impacts to model calibration (and in particular to the light penetration formulation); iv) the abstract should be adapted accordingly.

Response: Thank you very much for this constructive suggestion. We will revise it as suggested.

I think that the comparison with independent BGC-Argo floats is a valuable aspect of the manuscript, however, it would be interesting to know the spatial-temporal distances of the non-assimilated profiles with respect to the assimilated ones. Did they cover similar areas of the gulf of Mexico? And in the same period? In my opinion clarification on this aspect would help to better understand and comment the relatively small impact of Argo profiles assimilation when compared to the independent BGC-Argo data. Moreover, this could help also in commenting the differences between the two maps of Fig. 6 (are the differences mainly located close to assimilated Argo profiles?). Response:

Figure r1 shows positions of Argo profiles (gray dots) at each data assimilation cycle (e.g. 7 Jan 2015, the first update date in our data assimilative experiments) and BOEM profiles (colored squares) before the next one (e.g. from 7 Jan 2015 to 14 Jan 2015). The solid black circles represent areas within one localization radius (50km) from each Argo profile. Colors of squares represent the days of each BOEM profile after each data assimilation cycle. As shown in Figure r1, most of BOEM profiles are outside of one localization radius from the Argo profiles and therefore are barely updated by assimilating the Argo profile. Figure r2 shows the root-mean-square-difference (RMSD) of temperature from each BOEM profile between two data assimilative runs ( $RMSD = \sqrt{\frac{1}{n}\sum(DA_{Sat} - DA_{Argo})^2}$ ) The x-axis represents days of each BOEM profile after each data assimilation cycle and the y-axis represents distance to the nearest Argo

profile. In general, the RMSD between two data assimilative runs decreases with the distance but shows no significant decreasing trends with the days after update. This means that the differences induced by assimilating Argo profiles can be well sustained locally by model dynamical adjustments. The overall similarities between two data assimilative runs in Figure 4 can be explained to some extent by the large distances between BOEM and Argo profiles. However, it doesn't mean that increasing the localization radius necessarily improves the data assimilation performance. The current localization radius was determined by initial tests in Yu et al. (2019).

The differences in RMSE of surface chlorophyll between two data assimilative runs are shown in Figure r3. Positions of the assimilated Argo profiles are superimposed. The major differences in the two data-assimilative runs are mainly located around the Argo profiles. Reasons for why assimilating the Argo profiles can degrade surface chlorophyll are given in our response to the reviewer 1 and in the discussion of our original manuscript.

---

## Author Comment (AC3)

**Responses to Reviewer 3**

Below the complete reviewer comments are shown in black font along with detailed responses to each comment in blue font.

**Review:**

This study implements DEnKF-based data assimilation of satellite sea surface temperature, sea surface height and chlorophyll, and in situ temperature and salinity, with a physical-biogeochemical model of the Gulf of Mexico. The results are validated using the assimilated data plus profiles from five independent BGC-Argo floats, with a particular focus on subsurface biogeochemistry. Following the validation, a change was made to the light attenuation parameterisation to improve the fit to BGC-Argo chlorophyll.

The paper is well written and the experiments and results useful and well described. Subject to a few minor revisions, detailed below, I recommend publication in Ocean Science.

Response: We thank the reviewer for the constructive comments and suggestions which will be very helpful as we revise the manuscript.

P2 L41 - "discretion schemes" should presumably be "discretization schemes"? Response: We will correct it as suggested

P3 L65-70 - while less focus has definitely been given in the literature to validating the subsurface than the surface, it's not as rare as this paragraph would suggest. Some of the studies already referenced, and more besides, perform some validation of the subsurface, including on a point-to-point basis. See e.g. Fontana et al., 2013; Ford and Barciela, 2017; Mattern et al., 2017; Cossarini et al., 2019. The paragraph should be rephrased accordingly.

Response: We thank the reviewer to point out this. We will include these references and rephrase this paragraph into:

"The insufficient availability of subsurface and interior ocean biogeochemical observations is not only reflected in the immaturity of biogeochemical data assimilation but also its skill assessment. When compared with the surface, the subsurface has received less attention in skill assessments of biogeochemical data assimilation systems. Although there already have been studies which compared their vertical structures with in-situ observations and/or climatological datasets (e.g. Fontana et al., 2013; Ford and Barciela, 2017; Mattern et al., 2017; Ourmières et al., 2009; Teruzzi et al., 2014), these validations are often limited to low spatio-temporal resolution. The recent growth of autonomous observation systems, esp. BGC-Argo floats and gliders, make it possible to evaluate biogeochemical data assimilation systems below the surface in high resolution (e.g. Cossarini et al., 2019; Salon et al., 2019; Skákala et al., 2021; Verdy and Mazloff, 2017)."

**P3 L74 - I think "Garcon" should be "Garcon".**

Response: True. We will correct it in our revised manuscript.

**Is any transformation performed for chlorophyll to deal with non-Gaussianity?**

Response: In this study, we assimilate the actual chlorophyll concentrations. Based on our prior tests, assimilating log-chlorophyll can yield more improvements of the surface chlorophyll in coastal regions but degrade it in the deep ocean (with depth>1,000m). In addition, assimilating log-chlorophyll results in less improvements of subsurface biological properties, including the chlorophyll, phytoplankton, and POC. As this study focused on the subsurface biological properties in the deep ocean of the Gulf of Mexico, we decided to assimilate the actual chlorophyll. Also, there already have been successful examples which assimilated or updated the actually chlorophyll (e.g. Hu et al., 2012; Yu et al., 2018). Nevertheless, we have to acknowledge that assimilating the actual chlorophyll is suboptimal because of its non-Gaussian nature and will state this more clearly in the revised manuscript.

**Do the observation errors used account for representation error? Is this likely to affect results?**

Response: We didn't include representation errors in our study. It is also a common practice to only account for the instrument errors in data assimilation. The detailed influence of the representation errors need to be further investigated.

**P8 L04 - would additional metrics, even just bias and correlation coefficient, give additional information? Furthermore, chlorophyll and other biogeochemical variables are not normally distributed. Is RMSE appropriate here?**

Response: We will also report bias and correlation coefficient in our revised manuscript. For the usage of RMSE, we understand the reviewer's concerns that the RMSE may skew to high values. In previous applications (e.g. Ford and Barciela, 2017; Pradhan et al., 2020, 2019), the RMSE of log-chlorophyll has been also applied to put a same weight on the low values. However, this study in particular focus on the subsurface biological properties in the deep ocean (with depth > 1,000m) of the Gulf of Mexico and the major subsurface features, e.g. the deep chlorophyll maximum, are relatively high in their order of magnitude (e.g.  $\sim o$  (1E-1 mg m-3) of chlorophyll). In addition, the low concentrations (e.g.  $\sim o$  (1E-2 mg m-3) chlorophyll) are less ecologically important than the high concentrations.

**P10 L56 - can the uncertainties be quantified?**

Response: Yes. The uncertainty can be quantified by two methods, error propagation and validation with in-situ observations. The former method propagates errors from input to output with good knowledge of uncertainties from the input and model parameters. The latter one will calculate the root-mean-square difference (RMSD) and bias between the satellite estimates and match-up in-situ observations. Some satellite products (e.g. Ocean Colour CCI) can provide the pixel-by-pixel uncertainties. For this purpose, the uncertainty will be firstly calculated with respect to different optical water types and then extrapolated into each pixel as the weighted sum of uncertainties for each water type.

**P11 L87 - how was significance calculated?**

Response: In this study, the significant improvement is referred as the reduction in RMSE larger than or equal to 10% (please see the caption of Table 2 in our original manuscript). We will clarify this in our revised manuscript.

**P11 L93-94 - what method was used to calculate the old and new parameterisations?**

Response: Parameters of the original light attenuation scheme ( $Att=0.04+0.025 \times chl$ ) is based on previous studies (e.g. Fennel et al., 2011, 2006) and the alternative light attenuation parameterization ( $Att=0.027+0.075 \times chl^{1.2}$ ) is subjectively tuned based on the BGC-Argo floats. We will add some explanation for the two light parameterizations in our revised manuscript.

**Please comment further on why Fig. 9 shows limited improvement in the free run with the new parameterisation. Does this imply that the parameterisation is too tuned to the BGCArgo data, or that state-parameter estimation might be the best long-term approach?**

Response: The over-tuning means that a model reproduces observations through wrong mechanisms. However, in the Free\_alt run of our study, the physical circulation features, e.g. the position of Loop Current eddy, are wrong. We cannot expect the alternative parameterization to improve chlorophyll by compensating errors from physical circulations, which otherwise is a symptom of the over-tuning. In the DAsat\_alt run, the chlorophyll is improved as the data assimilation corrects physical circulation features, which indicates that our alternative parameterization is well calibrated.

**Fig. 1 - please state what the red square represents. Is this the model domain?**

Response: Yes, the red rectangle represents our model domain. We will clarify it in our revised manuscript.

Fig. 3 - please include units where appropriate. (a-e) should probably be (a,b,d,e). Response: We will revise it as suggested

**Fig. 7 - should the observations be included in a subplot here, as in Fig. 4?**

Response: We didn't show NO3 observations because NO3 is not measured by the BGC-Argo floats in the Gulf of Mexico. In this study, we estimated NO3 based on its climatological relationship with temperature (Figure S1 of our original manuscript which is replot as Figure r1 here) and compared it with the model results. We will add the estimated NO3 distributions in our revised manuscript. However, the estimated NO3 tend to be overestimated in the high temperature regions (Figure r1), which typically occurs within the euphotic layer. Therefore we used the unbiased root-mean-square-error to quantify the model-data misfit of NO3.

Figure r1 Empirical relationship of temperature-NO3 derived from the World Ocean Atlas in the Gulf of Mexico. Colors indicate the number of observations within each bin.

**Fig. 10 - what is meant by DCM increment? DCM is not one of the state variables included in the assimilation.**

Response: The definition of DCM increment is analogous to increment of other state variables. Specifically, we calculate the DCM depth based on chlorophyll profiles before (forecast) and after (analysis) update, and we define the changes due to update as the DCM increment. We will add a definition in our revised manuscript.

Fig. S3 does not appear to be referred to in the text. More generally, the figures in the supplement could reasonably be included in the main paper, but I appreciate the desire to limit the number of figures. Response: We will refer to this figure in our revised manuscript. In addition, as suggested by the reviewer 2, we will reorganize the manuscript to introduce the alternative light parameterization in a different way. We may include this figure into the manuscript.

**Reference**

[revised manuscript text omitted]

---

## Author Comment (AC4)

**Responses to Reviewer 4**

Below the complete reviewer comments are shown in black font along with detailed responses to each comment in blue font.

**Review:**

The study is investigated the ability of available satellite information on ocean surface physical and biological properties to constrain and improve simulated subsurface biogeochemistry in the Gulf of Mexico. The study also shows an example of using complementary Argo data. In this respect the paper nicely suits the frame of the journal. Generally, the manuscript is well structured and clearly written. The figures are of a good quality. I have got just minor comments (please see below) the authors might still want to consider before publishing the manuscript.

Response: We thank the reviewer for the constructive comments and suggestions which will be very helpful as we revise the manuscript.

Title: an edit is required "...assimilation of satellite physical and biological observations ...". It is a bit confusing: the title emphasizing the use/role of satellite information in improving subsurface biogeochemistry, however Argo data are also assimilated. Could the title reflect the use of Argo data? Response: We feel the title is appropriate as is and would like to keep it. Although we also assimilate Argo profiles in DAargo run, this is only core-Argo data (i.e. temperature and salinity) and not BGC Argo data. The key point of this study is to evaluate the impact of assimilating satellite observations on the subsurface distributions, especially of biological properties. Moreover, the additional improvements obtained by assimilating Argo profiles are limited.

**Specific comments:**

Line 15:16. reads as BGC-Argo data are also assimilated complementary to the satellite data. Reads a bit contradictory to the title or vice versa the title reads contradictory to the statement.

Response: The satellite data are assimilated into the coupled model and the BGC-Argo data are used to calibrate the biological model. We will revise this sentence into:

'This study investigates to what degree the assimilation of satellite observations in combination with a prior model calibration by sparse BGC Argo profiles can improve subsurface biogeochemical properties'

**P1, L18: "... into a three-dimensional biogeochemical model ..."**

Response: We will revise it as suggested. As also suggested by the reviewer 2, we will rephrase the lines 16-21 in our original manuscript into:

"..... The multivariate Deterministic Ensemble Kalman Filter (DEnKF) has been implemented to assimilate physical and biological observations into a three-dimensional coupled physical-biogeochemical model, of which the biogeochemical component has been calibrated by the BGC-Argo floats data for the Gulf of Mexico. Specifically, observations of sea surface height, sea surface temperature, and surface chlorophyll were assimilated, and profiles of both physical and biological variables were updated based on the surface information....."

**P2, L30-31: How was the tuning done? To a certain extent it is still a kind of assimilation of the information.**

Response: We agree. From a broad perspective, any practice to constrain a model by observations can be referred to as data assimilation. In this study, when we say data assimilation, we specifically refer to state estimation. We will state this clearly in the revied manuscript. Available BGC Argo float data are still insufficient for three-dimensional state estimation but can be used for *a prior* tuning of model. As a result, the well-tuned model can reproduce the key processes, e.g. the feedbacks between chlorophyll and light intensity in our study, and maintain the improvement obtained in update steps. The alternative light parameterization was tuned subjectively by BGC-Argo floats. However, the *a priori* tuning can be also done in other ways, such as parameter optimization.

P2, L41: "discretization and numerical schemes" instead of "discretion" Response: We will revise it as suggested.

P2, L45: "." is missing in the end of the sentence. Response: We will add this as suggested.

P2, L48: suggest to add "(e.g. Chla)" after "satellite data of ocean colour have been the major source of observations"

Response: We will revise it as suggested.

P2, L51: correct reference is Pradhan et al., 2020 Response: Thank you for pointing out this. We will correct it as suggested.

**P4, L85-86: are the mentioned five BGC-Argo floats really independent if used for the model calibration (model optimisation even though by "trails-and-error")?**

Response: We understand the reviewer's concern that the BGC-Argo float data are not fully independent because they have been used to optimize the biological parameters in Wang et al. (2020). Here the 'independent observations' refers to the unassimilated observations. We believe this is consistent with the common understanding in the data assimilation literature and therefore we will keep it as it is but we will point out clearly that the data has been used in prior tuning in the revised manuscript.

**P5, L31: "observational operator" instead of "measurement operator"**

Response: We will revise it as suggested.

**P6, L53-54: The specified (assumed) observational errors for SSH and SST are quite small, which could lead to overfit to the data with possible deterioration of the state for other model variables.**

Response: The observational errors of SSH and SST are based on previous studies (Song et al., 2016; Yu et al., 2018, 2019) and have been applied successfully in Yu et al. (2019). With respect to the overfitting issue, we compared model results below the surface with unassimilated Argo and BGC-Argo profiles. The subsurface temperature, salinity, and NO3 were largely improved. Other biological variables were also improved although the improvements were limited. These results give us some confidence that the observational errors are appropriate.

P7, L79-80: Inflation is normally introduced to account for uncertainties in approximation of model error (due discrepancies in the forcing or internal model parameter/parameterisations), which consequently alters the ensemble spread.

Response: We agree. We will revise it into:

'In addition, ensemble anomalies are inflated by 1.05 at each update step to account for the unrepresented model uncertainties (Anderson and Anderson, 1999).'

P7, L89-91: It would be nice to provide a reference to a study on model sensitivity to these particular parameters? A motivation and a reference to a procedure of parameter perturbation would support. Please also make it clearer whether the parameters are perturbed just to introduce more stochasticity to the system (e.g. Pradhan et a. 2019, 2020) or the data assimilation experiment considers also parameter estimation (Doron et al., 2011, Simon et al. 2015).

**Response:**

The sensitive biological parameters were selected by sensitivity tests. Specifically, we ran a 1D version of this model multiple times by incrementally perturbing one parameter at a time and setting other parameters unchanged (hereafter as **Test case**). Each parameter (p) was perturbed by multiplying factors ranging from 0.25 to 1.75 with an increment of 0.25. Then parameters were sorted based on their sensitivity

which was quantified by the normalized absolute differences from the unperturbed run (hereafter as **Base case**):

$$Q(y,p) = \frac{1}{m} \sum_{i=1}^{m} \frac{1}{n} \sum_{j=1}^{n} \frac{|y_{Base} - y_{Test}|}{y_{Base}}$$

where *m* is the number of parameter increments (here 7) and *n* is the number of base-test pairs including all 1D model grid cells throughout the whole simulation period. Based on the sensitivity tests, the most sensitive parameters were selected and perturbed in our data assimilation experiments. The details about the 1D model can be found in the section 3.3 of Wang et al. (2020). In our revised manuscript, we will add explanation about the choice of sensitive biological parameters.

In this study, the sensitive biological parameters were perturbed to introduce model uncertainties but were not updated. We will make it clearer in the revised manuscript.

P8, L1-3: Please consider rephrasing this sentence. The length of the state vector should not affect crucially the computational cast. Normally the time required for the analysis (independent on the length of the state vector) takes much less than the computational expenses required for running the ensemble itself. It is worth providing another argument for justification of the choice of model variables included to the state vector. Response: As suggested, we will rephrase this sentence into:

'Although the DEnKF can update all variables based on their cross-covariance, we limit updates to two physical variables (temperature and salinity) and four biological variables (nitrate, chlorophyll, phytoplankton, and zooplankton) which are key to the coupled physical-biogeochemical system. Other variables will be adjusted by model dynamics.'

P8, L4-12: It is worth showing both criteria RMSD and unbiased RMSD (+ additional? bias). In this case it would be clearer for the reader for which model variables the solution deviate systematically or randomly from the observations, whether the data assimilation allows to reduce bias (if any) or random part of the obtained differences between model and observations.

Response: Thank you for this suggestion. We will report the bias and correlation coefficient in our revised manuscript as suggested.

P9, L24: "This figure shows" or "This comparison shows" Response: We will revise as suggested.

Figure 2 could be slightly increased.

Response: We will revise as suggested.

Supplement, Figure S3: a legend or more detailed figure caption is required. Response: We will revise as suggested.

**Reference**

- Song, H., Edwards, C. A., Moore, A. M. and Fiechter, J.: Data assimilation in a coupled physicalbiogeochemical model of the California current system using an incremental lognormal 4-dimensional variational approach: Part 3—Assimilation in a realistic context using satellite and in situ observations, Ocean Modelling, 106, 159–172, doi:https://doi.org/10.1016/j.ocemod.2016.06.005, 2016.
- Wang, B., Fennel, K., Yu, L. and Gordon, C.: Assessing the value of biogeochemical Argo profiles versus ocean color observations for biogeochemical model optimization in the Gulf of Mexico, Biogeosciences, 17(15), 4059–4074, doi:10.5194/bg-17-4059-2020, 2020.
- Yu, L., Fennel, K., Bertino, L., Gharamti, M. El and Thompson, K. R.: Insights on multivariate updates of physical and biogeochemical ocean variables using an Ensemble Kalman Filter and an idealized model of upwelling, Ocean Modelling, 126, 13–28, doi:https://doi.org/10.1016/j.ocemod.2018.04.005, 2018.

Yu, L., Fennel, K., Wang, B., Laurent, A., Thompson, K. R. and Shay, L. K.: Evaluation of nonidentical versus identical twin approaches for observation impact assessments: an ensemble-Kalman-filterbased ocean assimilation application for the Gulf of Mexico, Ocean Science, 15(6), 1801–1814, doi:10.5194/os-15-1801-2019, 2019.